# Effects of Indonesian Shortfin Eel (*Anguilla bicolor*) By-Product Oil Supplementation on HOMA-IR and Lipid Profile in Obese Male Wistar Rats

**DOI:** 10.3390/nu15183904

**Published:** 2023-09-07

**Authors:** Ginna Megawati, Siti Shofiah Syahruddin, Winona Tjandra, Maya Kusumawati, Dewi Marhaeni Diah Herawati, Dida Achmad Gurnida, Ida Musfiroh

**Affiliations:** 1Doctoral Study Program, Faculty of Medicine, Universitas Padjadjaran, Bandung 40161, Indonesia; ginna@unpad.ac.id; 2Division of Medical Nutrition, Department of Public Health, Faculty of Medicine, Universitas Padjadjaran, Sumedang 45363, Indonesia; 3Medical Undergraduate Program, Faculty of Medicine, Universitas Padjadjaran, Sumedang 45363, Indonesia; 4Department of Internal Medicine, Faculty of Medicine, Hasan Sadikin Hospital, Universitas Padjadjaran, Bandung 40161, Indonesia; 5Department of Child Health, Faculty of Medicine, Hasan Sadikin Hospital, Universitas Padjadjaran, Bandung 40161, Indonesia; 6Department of Pharmaceutical Analysis dan Medicinal Chemistry, Faculty of Pharmacy, Universitas Padjadjaran, Sumedang 45363, Indonesia

**Keywords:** Indonesian shortfin eel by-product oil, fish oil supplementation, *n*-3 fatty acids, omega-3, HOMA-IR, lipid profiles, total cholesterol, LDL, HDL, triglyceride

## Abstract

The prevalence of people being overweight and obese has increased globally over the past decades. The use of omega-3 fatty acids—a compound usually primarily found in fish oil—has been known to improve the metabolic profile of obese patients. As the demand for eels increases, the number of waste products from the eels increases and creates environmental problems. This study was conducted to investigate the effect of a newly discovered Indonesian Shortfin eel by-product oil supplementation on the Homeostasis Model Assessment-Estimated Insulin Resistance (HOMA-IR) and lipid profiles of obese male (Lee index ≥ 0.3) Wistar rats (*Rattus norvegicus*). The oil was extracted from waste products (heads). Fifteen obese rats were divided into three groups and were administered NaCl (C), commercial fish oil (CO), and Indonesian shortfin eel by-product oil (EO). All groups had statistically significant differences in total cholesterol, LDL, and triglyceride levels (*p* < 0.05). The CO and EO group showed a significant decrease in total cholesterol, LDL, and triglyceride after treatment. However, no significant difference was found in HDL levels and HOMA-IR. The supplementation of Indonesian shortfin eel by-product oil significantly improved lipid profile while effectively mitigating environmental challenges.

## 1. Introduction

According to the World Health Organization (WHO), in 2016, more than 1.9 billion adults worldwide were overweight, with over 650 million obese [1]. In 2019, approximately 5 million deaths worldwide were attributed to obesity-related causes. An analytical study of the Global Burden of Disease Study 2019 has predicted an anticipated 42.7% surge in mortality linked to obesity from 2020 to 2030 [2]. Correspondingly, a study revealed a twofold increase in comorbidity prevalence, including diabetes, hypertension, dyslipidemia, and osteoarthritis, among individuals who were overweight or obese [3]. This trend is notably mirrored in the prevalence of diabetes. In 2017, the global impact of type 2 diabetes was profound, affecting an estimated 462 million individuals, constituting approximately 6.28% of the global population. The gravity of the issue was evident, as diabetes alone accounted for over a million deaths in that same year, making it the ninth leading cause of mortality [4]. With all these observations, the further development of effective interventions is imperative. The focus of the current management of obesity has been shifted from weight loss to the prevention and control of obesity-related comorbidities [5]. Furthermore, healthy dietary interventions, serving as a cornerstone of lifestyle modifications, have been advocated for effective diabetes management. These interventions play a pivotal role in achieving personalized targets for glycemic control, blood pressure regulation, and lipid profiles [6,7].

Until recently, the use of omega-3 fatty acids has not shown a consistent benefit for weight loss, but has shown an improvement in the metabolic profile of obese patients [8]. Among the metabolic effects are improved lipid profiles, such as total cholesterol, low-density lipoprotein (LDL), high-density lipoprotein (HDL), and triglyceride levels [9,10,11,12]. In addition, omega-3 has been proven to improve insulin sensitivity, the beta-oxidation of fatty acids in the liver, muscles, and the heart, and suppress inflammation [13,14,15] in subjects who are already obese. Consequently, it can prevent the emergence of comorbidities, such as insulin resistance, dyslipidemia, and hypertension, collectively called metabolic syndrome [16,17,18,19]. Insulin resistance can be estimated using the Homeostasis Model Assessment-Estimated Insulin Resistance (HOMA-IR) parameters with a cut-off value of two to be considered insulin resistant, according to most sources, including in rats [20,21,22,23].

Fish oil is widely known as a dietary source with a high concentration of omega-3 eicosapentaenoic acid (EPA) and docosahexaenoic acid (DHA) [24,25]. Eels (*Anguilla* spp.) are catadromous fish, meaning that they live in freshwater but migrate to seawater to breed, and have long been known to have a rich omega-3 content [26,27,28]. Several studies have indicated that Indonesian shortfin eel (*Anguilla bicolor*) oil provides an anti-inflammation effect and triglyceride-lowering feature in hyperlipidemic rats [29,30]. Eel has been one of the highest fishery commodities in the international market [31]. In 2020, Indonesia was the biggest eel exporter, with an export volume of over 10,000 tons per year or about 25% of world eel exports [31]. With the high demand for eels, the number of waste products increases, creating environmental problems that must be compensated for by reducing, reusing, or recycling [32]. Based on this environmental concern, using eel by-product oil high in omega-3 becomes a potential solution to both ameliorating comorbidities in obese individuals and reducing environmental waste.

This study aimed to investigate the potential health benefits of a recently developed Indonesian shortfin eel oil derived from waste products (heads) on diet-induced-obese male Wistar rats. After treatment, the HOMA-IR, total cholesterol, LDL, HDL, and triglyceride levels were analyzed to indicate the oil’s efficacy. Weekly measurements of body weight were obtained. It is anticipated that the results of this pre-clinical study will serve as a foundation for future research on the effect of eel by-product oil on obesity at the clinical stage.

## 2. Materials and Methods

This study was reported according to the latest ARRIVE (Animal Research: Reporting of In Vivo Experiments) guidelines [33].

### 2.1. Animals and Diets

Twenty-seven male Wistar rats (*Rattus norvegicus*) were obtained from PT. Biofarma, located in Bandung, West Java, Indonesia, at 10–12 weeks. They were maintained under control temperature (temperature), humidity, and airflow condition, with a fixed 12:12 h light-dark cycle. After one week of acclimation, the rats were fed to achieve obesity (Lee index ≥ 0.3, calculated with [3 square root body weight (g)/naso-anal length (cm)] × 1000) with diet-induced obesity (DIO) method for 17 weeks. The DIO method used a high-fat diet from the Prospet brand with 37.44% fat (5bb) analyzed with the 1922 *Standar Nasional Indonesia* (SNI) analysis method. The initial age of rats before treatment was 28–30 weeks, and the baseline characteristics are presented in Table 1. The animals were caged based on their previous DIO method and treatment (three animals per cage). The sample size was calculated using Mead’s equation before the data collection started [34]. The formula was as follows: E = N − B with E = error degrees of freedom, N = number of samples, and T = number of treatments; E should be between 10 and 20. A total of 5–7 rats per group were considered necessary. Hence, the minimum total number of rats needed for the study was 15.

The obese rats were randomly divided into three groups: (1) control group (C): animals fed with NaCl (0.9%) with the dose of 0.5–1.0 mL/rat/day; (2) commercial fish oil group (CO): animal fed with commercial fish oil (Blackmores Fish Oil, Alpha Lab., New Zealand, imported by PT. Kalbe Blackmores Nutrition, Indonesia); and (3) Indonesian shortfin eel by-product oil group (EO): animal fed with Indonesian shortfin eel by-product oil. The animals were included in the study if they met the criteria of obese rats based on the Lee index (≥0.3) and were in healthy condition. The animals were excluded if the rat experienced a weight loss of more than 20% during the period of obesity adaptation; there were invalid data in any of the lipid profiles (total cholesterol, LDL, HDL, or triglyceride) due to undetected results (outside measurement range); the results were considered outliers based on a statistical method (histogram). Based on these criteria, several animals were not included in the analysis. The details are as follows: 12 rats were not included, with a total of 15 rats left for analysis (*n* C = 5, *n* CO = 5, *n* EO = 5). The number of rats in each group differed from the sample size calculation due to the exclusion.

The provision of food and water was ad libitum and used the previous DIO feeds. The commercial and Indonesian shortfin eel by-product oil treatments were given orally using a cannula needle. The treatment took place for four weeks based on previous studies [35,36]. Body weights were measured weekly. At the end of the trial, the rats were sacrificed by placing them in a container containing a lethal dose of ether, and both their fat mass and liver biopsy samples were collected. Confounder factors are not controlled. The treatment was conducted with a blinding method to minimize bias: the laboratory assistant carried out the treatment and outcome measurement, the results of which were unknown to the authors.

The ANOVA analysis followed by Tukey’s post hoc test revealed significant statistical differences (*p* < 0.05) in food intake and total caloric intake between the control group and the group receiving fish oil. However, no statistically significant differences were observed between the group receiving commercial fish oil (CO) and eel by-product oil (EO).

### 2.2. Oil Preparation

Preparation of Indonesian shortfin eel head oil (by-product) was carried out according to the International Fish Oil Standard (IFOS). Neutralization was carried out using neutralizing alkali (NaOH). The results of the oxidation test parameters obtained Free Fatty Acid (FFA) 1.12%, Peroxide Value (PV) (2.64 meq/kg), Anisidin Value (AnV) 12.47 meq/kg and Total Oxidation (TOTOX) 17.85 meq/kg. All these parameters meet the set standards.

### 2.3. Oil Composition

The Indonesian shortfin eel by-product and commercial oil fatty acids composition are presented in Table 2. Using a preceding study, we successfully prepared the Indonesian shortfin eel by-product oil and subjected it to various tests, including exploration, fish oil quality, heavy metal contamination, in silico, and fatty acid content tests. The results obtained from these tests unequivocally met the established standards, demonstrating the high quality of the eel by-product oil. 

Table 2 illustrates that the composition of SFA and PUFA in eel by-product oil has almost the same percentage as commercial oil. However, eel by-product oil has a higher composition of MUFA (42.13%) than commercial oil (24.30%). There is a difference in the EPA/DHA composition between the two oils. Indonesian shortfin eel by-product oil has a higher composition of DHA (17.45%) than its EPA and the DHA in commercial oil (12.4%). Commercial oil has a higher composition of EPA compared to its DHA.

### 2.4. Oil Dose Determination

Humans’ omega-3 fatty acids dose (60 kg) was 600 mg/day. Those doses were converted to *Rattus norvegicus*, resulting in the 1.72–2.15 mg/kg body weight. This dose was used to determine the dose of commercial and eel by-product oil given to the rats. The oil dose was adjusted to the omega-3 fatty acids content in commercial and Indonesian shortfin eel by-product oil. A total of 1000 mg of commercial fish oil contains 300 mg of omega-3 PUFAs, 180 mg of EPA, and 120 mg of DHA. Meanwhile, the Indonesian shortfin eel by-product oil contains 95.14 mg of omega-3 PUFAs. Based on these figures, the equivalent dosages for eel by-product oil and commercial fish oil were calculated. The fish oil doses for the commercial fish oil group (CO) were 0.4–0.5 mL/rat/day and 1.26–1.57 mL/rat/day for the Indonesian shortfin eel by-product oil group (EO).

### 2.5. Metabolic Parameters and Blood Sampling

Blood was taken twice according to a conscious model: (1) before treatment, blood was drawn from the tail with a 26G needle; (2) after treatment, blood was drawn from the heart with a 23G needle and 3 mL syringe. The blood was placed in an Ethylene Diamine Tetra-acetic Acid (EDTA) container. The blood was centrifuged (3000 rpm, 15 min) at room temperature to separate the serum. The serum was used to measure metabolic parameters. Blood triglyceride levels were determined by the colorimetric enzymatic method using glycerol-3-phosphate-oxidase (GPO) [15]. The determination of cholesterol is based on a direct examination of Cholesterol Oxidase-Peroxidase Aminoantipyrine (CHOD-PAP)/enzymatic photometric test [37]. A photometer measured the results. HDL and VLDL remained in the supernatant after centrifugation and were measured enzymatically by the CHOD-PAP method [38]. The HDL levels were measured using precipitation reagents as chylomicrons, and VLDL and LDL were precipitated by adding phosphotungstic acid and magnesium ions to the sample. The HDL in the supernatant is determined enzymatically according to the CHOD-PAP method [39]. The analysis of LDL cholesterol was performed on indirectly measured LDL cholesterol levels using the Friedewaldetal formula, which involves subtracting total cholesterol by HDL and one-fifth of triglycerides. DiaSys Diagnostic Systems GmbH manufactured all measurement tests for lipid profiles.

### 2.6. HOMA-IR Calculation

To estimate insulin sensitivity, fasting plasma glucose and insulin (6–12 h fast) were measured to calculate the HOMA-IR. Blood samples were taken before interventions from the tail vein and after interventions from the heart. Glucose concentration was measured using the Glucose Oxidase-Peroxidase Aminoantipyrin (GOD-PAP) Enzymatic Photometric Test (DiaSys Diagnostic System GmbH, Holzheim, Germany). Meanwhile, insulin concentration was measured using the Enzyme-linked Immunosorbent Assay (ELISA) method (ABclonal Technology Co., Ltd., Wuhan, China). HOMA2 Calculator application from the University of Oxford was used to calculate HOMA-IR, 173 with a range of measurement: plasma glucose 3.5 to 25.0 mmol/L; plasma insulin 20 to 400,174 pmol/L; plasma specific insulin 20 to 300 pmol/L; plasma C-peptide 0.2 to 3.5 nmol/L. The equation for calculating HOMA-IR was as follows: HOMA-IR = [fasting plasma insulin 176 (ng/mL) × fasting plasma glucose (mg/dL)]/405.

### 2.7. Statistical Analysis

Results are expressed as means (standard deviation). Before the analysis, the data were tested by normality (Shapiro–Wilk) and homogeneity tests (Levene Statistic) to determine whether the data met the assumptions of the statistical approach. For those who met the criteria (the distribution was normal and the data were homogenous), the significance between groups was determined by one-way ANOVA followed by the inspection of all differences between pairs of means by the Post Hoc Tukey test. All statistical analysis was conducted using IBM SPSS Statistics 26.0 software. Differences with *p*-values < 0.05 were considered significant.

### 2.8. Ethical Consideration

The ethical principles for conducting animal studies consist of 3Rs (replacement, reduction, and refinement) and 5Fs as follows: (1) freedom from hunger and thirst; (2) freedom from discomfort; (3) freedom from pain, injury, and disease; (4) freedom to express normal behavior; and (5) freedom from fear and distress [40,41]. All ethical principles were applied to all the rats during this study. This study has received ethical approval from the Research Ethics Committee Universitas Padjadjaran, No. 481/UN6.KEP/EC/2022.

## 3. Results

### 3.1. Changes in HOMA-IR Levels before and after Interventions

The mean value of fasting plasma insulin and fasting plasma glucose were presented in Table 1, while the HOMA-IR values before and after interventions and the differences (delta) were presented in Table 3.

Based on the ANOVA test results, there was no statistically significant difference in the delta of HOMA-IR values between the three groups (F = 0.008, *p* = 0.99). There was an increase in the HOMA-IR values of the three groups, with the slightest increase occurring in the EO group, even though it was not statistically significant.

### 3.2. Changes in Lipid Profile Levels before and after Interventions

The changes (delta) in LDL levels in the obesity model rats treated with fish oil can be observed in Table 4. According to the ANOVA test, significant statistical differences were found in the changes in LDL levels among the obesity model rats in the control (C), commercial fish oil (CO), and eel by-product oil (EO) groups (F = 18.296, *p* = 0.001). Post hoc analysis using the Tukey method revealed significant differences in the changes in LDL levels between the EO and C groups (*p* = 0.001) and between the CO and C groups (*p* = 0.001). There were no statistically significant differences in the changes in LDL levels between the CO and EO groups (*p* = 0.991).

Based on the ANOVA test, no significant differences were found in the changes in HDL levels among the obesity model rats in the C, CO, and EO groups (F = 0.983, *p* = 0.402). All groups showed a decrease in HDL levels. A minor decrease was observed in the group treated by eel by-product oil, as depicted in Table 4.

The ANOVA test indicated significant statistical differences in the changes in triglyceride levels among the obesity model rats in the C, CO, and EO groups (F = 7.918, *p* = 0.006). Post hoc analysis using the Tukey method further revealed significant variations in the changes in triglyceride levels between the EO and C groups (*p* = 0.012) and between the CO and C groups (*p* = 0.013). However, no statistically significant differences were observed in the changes in triglyceride levels between the EO and CO groups (*p* = 1.00).

## 4. Discussion

Fish oil has been found to have many benefits for improving metabolism among those with obesity, with various studies pointing to its ability to modulate metabolic pathways in critical metabolic organs such as adipose tissue, the liver, and skeletal muscle [42,43,44]. The primary purpose of this study was to investigate the effects of the newly discovered Indonesian shortfin eel by-product oil developed from waste products, which is high in omega-3, on several metabolic parameters: HOMA-IR and lipid profile.

### 4.1. The Effects of Indonesian Shortfin Eel By-Product Oil Supplementation on Insulin Resistance in Male Obese Wistar Rats

In obesity, the amount of fat that exceeds the storage capacity of adipose tissue in the body will trigger hypertrophy of the tissue. Enlarged adipose tissue undergoes several abnormal changes, including inflammation due to hypoxia and necrosis and decreased release of adiponectin, which plays a role in increasing insulin signaling, while the number of pro-inflammatory cytokines (interleukin-6, resistin, retinol-binding protein 4, tumor necrosis factor-alpha), which play a role in inhibiting insulin signaling, increases [45,46], and increased lipolysis occurs, which will then lead to an increase in free fatty acids. This condition will trigger the deposition of free fatty acids in other non-adipose tissues, such as the liver, muscle, heart, and pancreas, referred to as ectopic lipid storage. Prolonged/chronic inflammation, as well as lipotoxicity, will play a major role in decreasing insulin sensitivity, causing a condition called insulin resistance in both adipose and non-adipose tissues. These will eventually impair the function of those organs, leading to metabolic syndrome and various other complications that jeopardize health [8,45].

In this study, there was no statistically significant difference in the delta of HOMA-IR values between the three groups (*p* > 0.5) (Table 3). There was an increase in the HOMA-IR values of the three groups, with the smallest increase occurring in the EO group, even though it was not statistically significant.

Our results were contradictory to the previous studies. It is well established that omega-3 supplementation gives many health benefits, including a protective effect against excessive fat accumulation in the body, and is beneficial in reducing the risk of obesity, as proven by many experiments on various strains of rats [47,48,49,50]. Previously published studies also revealed that omega-3 can improve insulin sensitivity [15,16] in subjects who were already obese. Consequently, it can potentially prevent the emergence of comorbidities, such as metabolic syndrome, which includes insulin resistance [16,51].

Studies in male Wistar rats suggest that the group receiving food high in omega-3 fatty acids showed less accumulation of visceral/epididymal fat at the end of the study compared to the other groups [47,48]. Similar results were obtained by the other studies carried out in different strains of rats, Fisher 344 and C57BL/6 [50]. Another animal study in male C57BL mice indicated that high-fat diet (HFD)-induced-obesity mice experienced weight loss and improved HOMA-IR levels when HFD was substituted with a high omega-3 fatty acids diet [18,20,21]. A study in male Wistar rats also indicated that the consumption of omega-3 fatty acids in high-fructose diet-induced obesity rats showed improved insulin sensitivity. Hence, it can be concluded that in addition to their protective effect against excessive adiposity in rats, omega-3 fatty acids also have the potential to decrease weight and prevent insulin resistance in subjects who are already obese [18,51].

The fatty acid composition of Indonesian shortfin eel by-product oil contains high MUFA (42.13%) and SFA (26.79%). Previous studies showed that high SFA will induce the lipid-mediated inhibition of insulin signaling and lipid-induced stimulation of hepatic mitochondrial activity, leading to increased hepatic gluconeogenesis; consequently, these will prompt insulin resistance in the liver and other tissues [52].

Meanwhile, the role of MUFA in insulin resistance has yet to be conclusively determined. In other studies, MUFA effectively reduced glycosylated hemoglobin (HbA1c) and improved HOMA-IR levels in overweight or obese patients [53,54]. Although most evidence showed its benefit in insulin resistance, some studies suggest the opposite. MUFA induced insulin resistance in the liver and skeletal muscle by increasing serine phosphorylation of the β-subunit of the insulin receptor substrate-1 (IRS-1), inhibiting insulin signaling in the insulin cascade, and promoting insulin resistance. This was reflected by 61% higher rates of hepatic gluconeogenesis and a 25% decrease in whole-body insulin sensitivity. This mainly reflects skeletal muscle insulin action in a study involving glucose-tolerant volunteers. Besides that, it was shown that diets high in oleic acid (olive oil) and MUFA had no particular protective effect against the emergence of insulin resistance [53,55].

The difference in EPA and DHA levels between Indonesian Shortfin eel by-products and commercial fish oil still requires further exploration of its role in insulin resistance. In this study, there was no significant difference between the two groups that received fish oil treatment (*p* > 0.05) (Table 3).

Diets high in fatty acids, mainly EPA and DHA, can prevent the development of insulin resistance in rats. However, the definite individual roles of EPA and DHA on insulin resistance were poorly established because they were mostly used as a mixture. Previous studies in C57BL/6J mice fed high fat and high sucrose (HF-HS) diets and Ob/Ob mice indicated that EPA could limit adipose cell hypertrophy in visceral adipose tissue and fat mass accumulation in the early stage of weight gain, as well as preserve glucose homeostasis and insulin sensitivity in an obesogenic environment, leading to the prevention of further metabolic disturbances in mice. Besides that, EPA could inhibit two master transcription factors for adipocyte differentiation, peroxisome proliferator-activated receptor γ (PPAR-γ) and enhancer-binding protein (C/EBP) β, reducing adipogenesis while DHA demonstrates an opposite result, namely promoting fat mass accumulation [19,45,56].

DHA did not prevent fat mass accumulation or preserve glucose homeostasis. During their differentiation process, DHA increases leptin expression in the 3T3-L1 preadipocyte cell line. Nevertheless, there was another study conducted directly on 3T3-L1 preadipocytes, which postulated that DHA, compared with EPA, led to a greater increase in the secretion of adiponectin, important in increasing insulin signaling when administered at low concentration in 3T3-L1 preadipocytes [19,45,46,56].

The role of omega-3 fatty acids is as a supplement, not a drug; hence, it must be integrated with lifestyle modification to obtain the maximum result, including preventing metabolic syndrome [45,56]. Even though our findings contradict the previous studies, we believe there is still potential in using eel waste (by-products) as a source of omega-3 fatty acids.

### 4.2. The Effects of Indonesian Shortfin Eel By-Product Oil Supplementation on Lipid Profile in Male Obese Wistar Rats

This study revealed significant statistical differences (*p* < 0.05) in total cholesterol, LDL, and triglyceride levels among the groups receiving fish oil supplementation, namely commercial (CO) and eel by-product (EO) oil, in comparison to the control (C) group. Nevertheless, no significant distinctions were observed between the CO and EO groups concerning the aforementioned plasma lipid parameters (*p* > 0.05), as indicated in Table 4. Although there were no statistically significant disparities in plasma HDL levels among the three treatment groups, the mean delta HDL levels in the MS group exhibited the least reduction in HDL levels compared to the other groups.

The observed minimal decline in plasma HDL levels within the EO group, as compared to both the C and CO groups, highlights the potential efficacy of eel by-product fish oil. A recent study shows omega-3 intake elevated HDL levels in Wistar male rats [57]. These findings are consistent with previous studies that have demonstrated a reduction in cholesterol and triglyceride levels in a rat model subjected to a high-fat diet (HFD), wherein the administration of eel by-product fish oil outperformed the control group.

Another study [58] reported that fish oil rich in omega-3 PUFAs increased the plasma HDL of female Wistar rats after 21 days of treatment. The molecular mechanism underlying the HDL-raising effect of omega-3 was reported to be due to the decrease in the cholesteryl ester transfer protein (CETP) activity, which transfers the cholesterol esters from HDL to the VLDL and LDL that reduces the amount of HDL in the plasma [59,60].

The lipid profiles, including cholesterol, LDL, HDL, and triglycerides, have been considered in most studies about fish oil and its potential effect on obesity and metabolic syndrome [44]. Many experiments indicate the role of fish oil supplementation in ameliorating cholesterol, LDL, HDL, and triglyceride levels in rats who were obese and/or had metabolic syndrome [35,44,61,62,63,64,65,66,67,68,69,70,71]. A four-week fish oil treatment in male obese Wistar rats results in lower triglyceride, LDL, and total cholesterol serum levels [35]. A study by Chiuet et al. observed that HFD-fed rats supplemented with 5% fish oil showed a significant decrease in total cholesterol, LDL, and triglyceride levels compared to those without fish oil [61]. Using male Wistar rats as animal models, Pighiet et al. reported that the triglyceride levels were lower in the group supplemented by fish oil than those observed in the same diet without supplementation [68]. The numerous benefits of fish oil supplementation proved by these studies were confirmed by the evident decrease (*p* < 0.05) in total cholesterol and triglyceride levels in rats treated with commercial oil (CO) in our study compared to the control (C) group. The role of fish oil in ameliorating lipid profiles may be attributed to its omega-3 content.

Numerous mechanisms by which omega-3 fatty acids improve lipid profiles have been proposed related to transcription factors and enzymes involved in lipid metabolism. One of the two families of transcription factors regulating adipocyte differentiation is the peroxisome proliferator-activated receptor (PPAR) family [72]. Omega-3 PUFAs are natural ligands of this family of transcription factors. They have been known for their ability to activate PPAR α and PPAR γ, which are mainly expressed in the liver and adipose tissue, respectively [44,72]. The activation of these receptors promotes gene expression in lipid metabolism [72]. PPAR α promotes fatty acid oxidation within the hepatocytes, one of the methods of its elimination from the liver [72,73]. This leads to decreased substances needed for triglyceride biosynthesis, lowering triglyceride levels. Meanwhile, by stimulating several genes involved in adipogenesis, PPAR γ induces triglyceride storage, thereby decreasing the release of free fatty acids used for triglyceride synthesis in the liver [39].

Furthermore, various genes involved in fatty acid and cholesterol metabolism in the liver are regulated by these transcription factors, SRBPs [74]. It was observed that fish oil down-regulates the sterol regulatory element-binding protein 1 (SRBP1) by quickening the degradation of its mRNA, along with the decrease in mRNAs of cholesterologenic and lipogenic enzymes’ expression [74,75,76]. In addition, unsaturated fatty acids block the activation of SRBP-1c expression by antagonizing the ligand-dependent activation of the liver X-activated receptor (LXR), one of the factors that selectively regulates the transcription of SRBP-1c [74,77]. Consequently, these combined effects could reduce plasma triglyceride levels [74].

Additionally, omega-3 fatty acids’ triglyceride-lowering properties are due to their ability to decrease the synthesis of triglyceride by inhibiting diacylglycerol acetyl-transferase (DGAT), an enzyme involved in the formation of triglyceride, acetyl-CoA carboxylase (ACC), involved in free fatty acids’ biosynthesis, and increasing the beta-oxidation of fatty acids; all of these happen in the hepatocyte. Another possible mechanism is the inhibition of hormone-sensitive lipase (HSL) in adipose tissue, thus inhibiting fat lipolysis and decreasing the release of free fatty acids used for triglyceride synthesis [11].

The mechanism of fish oil agents for lowering LDL, HDL, and total cholesterol was closely related to triglyceride. As portions of hepatic triglyceride are secreted by very-low-density lipoproteins (VLDLs), a decline in triglyceride levels in the hepatocytes causes a decrease in VLDL production [78]. VLDL will then be converted into its intermediate-density lipoprotein (IDL) as it releases its content into tissues. With the help of Hepatic Triglyceride Lipase (HTGL), IDL is altered into LDL [78,79,80]. Consequently, reducing VLDL production leads to a decline in plasma LDL levels, showing the interrelation between triglyceride and LDL. On the other hand, HDL plays a significant role in reverse cholesterol transport (RCT), a mechanism in which the excess cholesterol is taken from peripheral tissues into the liver to be redistributed or excreted, showing its involvement in lipid metabolism [79,81].

The attempt to use fish by-product oil in animal trials to observe its potential anti-obesity properties still needs to be carried out. A study by Daidj et al. utilized a sardine by-product (head, scales, and viscera) oil. It showed a significant decrease in serum total cholesterol, LDL, and triglycerides in obese rats treated with by-product oil compared to fillet oil. These results might be due to the difference in omega-3/omega-6 ratio between the fillet and by-product oil: the ratio in the by-product oil was higher than in the fillet oil [82]. Consequently, the lipid-lowering effects of sardine by-product oil were more notable, as has been discussed before regarding the omega-3/omega-6 ratio. However, compared to several examples of other fish by-product oil, the composition of the oil used in the EO group is still relatively better, as shown in Table 2 [83,84,85,86].

Further research with oil preparation is needed, as the EO group required two–three times as much oil (by volume) per day as the CO group, given the potential confounding effects of these results in metabolic parameters. Longer intervention periods were also considered to explore the effects of eel by-product fish oil supplementation on dyslipidemia cases, considering its high content of monounsaturated fatty acids (MUFA) and saturated fatty acids (SFA), to obtain optimal potential regarding lipid metabolism and profile. It is well-known that SFA can elevate LDL cholesterol levels; thus, the intake of SFA has long been recommended to be limited in the diet [87,88,89]. The presence of SFA in eel by-product fish oil may overshadow the potential benefits derived from its omega-3 fatty acid content [90].

## 5. Conclusions

The administration of fish oil supplementation in this study revealed improvements in lipid profile parameters, including total cholesterol, LDL, and triglycerides, as compared to the control group, albeit without statistically significant differences observed between the fish oil group (CO) and the supplementation group (EO). The minimal decline of HDL levels in the EO group demonstrates the potential benefits of the newly discovered eel by-product oil to ameliorate metabolic parameters. Although it has no statistical significance, that the least decrease in HOMA-IR occurred in the EO group also indicates the potential benefit of eel by-product oil for controlling insulin resistance in obesity. Overall, the newly discovered Indonesian shortfin eel by-product oil has a promising benefit in improving lipid profile and insulin resistance. Research with a longer treatment time is needed to see a more tangible effect on the HOMA-IR and HDL parameters. At the same time, it also holds promise for reducing environmental problems, as it was recycled from waste products.

## Figures and Tables

**Table 1 nutrients-15-03904-t001:** Baseline characteristics of male obese Wistar rats.

Characteristics	C Group (*n* = 5)	CO Group (*n* = 5)	EO Group (*n* = 5)	*p*-Value
Weight (g)				
Before treatment	223.60 (40.15)	258.40 (39.59)	239.00 (33.33)	0.300
After treatment	234.20 (40.28)	265.60 (49.56)	241.80 (19.57)	0.430
Food intake (g)	15.41 (1.99)	12.69 (2.33)	11.82 (1.66)	0.004 *
Total caloric intake (Kcal) **	90.15 (11.67)	79.65 (13.66)	74.57 (9.74)	0.003 *
Total fat tissue weight (g)	3.82 (1.63)	3.07 (1.03)	3.29 (0.83)	0.620
Liver weight (g)	9.77 (0.70)	10.81 (1.33)	10.27 (0.70)	0.260
Waist circumference	15.28 (0.49)	15.66 (1.56)	15.400 (1.04)	0.860
Insulin (µIU/mL)				
Before treatment	5.45 (1.31)	5.23 (0.56)	5.37 (0.57)	0.920
After treatment	5.52 (0.64)	6.12 (0.91)	5.50 (0.49)	0.320
Glucose (mg/dL)				
Before treatment	90.92 (30.30)	112.68 (29.13)	105.45 (39.19)	0.580
After treatment	149.13 (34.67)	150.00 (27.40)	158.06 (29.28)	0.880

Values are expressed in mean (SD). C: control; CO: commercial fish oil; EO: Indonesian shortfin eel by-product oil. * *p* < 0.05 with one-way ANOVA test. ** Total caloric intake = energy intake from high-fat diet (HFD) + energy from fish oil for the CO and EO groups (300 mg = 5.4 Kcal).

**Table 2 nutrients-15-03904-t002:** Fatty acids composition in oil.

Fatty Acids	Indonesian Shortfin Eel By-Product Oil	Commercial Fish Oil
Lauric acid	0.09	n/a
Myristic acid	1.62	7.5
Palmitate	18.80	15.7
Stearic acid	6.27	3.1
Total SFA	26.79	26.3
Myristoleic acid	0.09	n/a
Palmitoleic acid	5.62	8.7
Oleic acid	33.90	8.5
11-Octadecenoic acid	2.52	7.1
Total MUFA	42.13	24.3
Linoleic acid	1.63	3.5
Arachidonate acid	6.10	1.1
Eicosanoic acid	0.23	1.07
EPA	5.58	17.5
DHA	17.45	12.4
Total PUFA	31.09	35.57
Omega-3	23.13	29.9
Omega-6	7.96	5.67

Composition of fatty acids (as % of total fatty acids). SFA: saturated fatty acids; MUFA: monounsaturated fatty acids; PUFA: Polyunsaturated Fatty Acids; EPA: eicosapentaenoic acid; DHA: docosapentaenoic acid; n/a: not available.

**Table 3 nutrients-15-03904-t003:** Changes (delta) in HOMA-IR levels of male obese Wistar rats.

Interventions	*n*	HOMA-IR			*F*	*p*-Value *
		Before	After	Differences		
C	5	1.22 (0.220)	2.01 (0.193)	0.79 (0.385)	0.008	0.99
CO	5	1.47 (0.223)	2.31 (0.336)	0.83 (0.478)		
EO	5	1.38 (0.213)	2.15 (0.227)	0.77 (0.189)		

Results are expressed as mean (standard error) for each group. C: control; CO: commercial fish oil; EO: Indonesian shortfin eel by-product oil. * One-way ANOVA Test; Shapiro–Wilk test: *p* = 0.0673; Levene test: F = 2.00, *p* = 0.14.

**Table 4 nutrients-15-03904-t004:** The plasma levels of total cholesterol, LDL, HDL, and triglyceride levels in obese rats fed with different treatments for four weeks.

Treatment	*n*	Before (mg/dL)	After	Delta	*F*	*p*-Value *
Total Cholesterol		Mean (SD)			23.83	0.001 *
C	5	145 (11.328)	167.90 (21.219)	22.26 (16.00)		
CO	5	182.05 (83.253)	105.05 (46.552)	−77.00 (38.84)		
EO	5	155.64 (8.485)	87.48 (9.925)	−68.97 (11.60)		
LDL		Mean (SD)			18.296	0.001 **
C	5	32.59 (11.099)	91.13 (28.056)	60.53 (25.935)		
CO	5	82.23 (67.716)	47.90 (33.164)	−34.33 (35.517)		
EO	5	54.89 (16.354)	22.82 (7.776)	−32.07 (21.631)		
HDL		Mean (SD)			0.983	0.402
C	5	100.65 (13.079)	56.74 (14.716)	−43.91 (23.874)		
CO	5	79.66 (14.439)	45.28 (10.319)	−34.38 (15.827)		
EO	5	77.60 (23.805)	49.85 (11.795)	−27.75 (13.660)		
Triglyceride		Mean (SD)			7.918	0.006 **
C	5	61.99 (36.007)	90.21 (26.134)	28.21 (15.500)		
CO	5	100.800 (76.757)	59.29 (33.480)	−41.50 (44.957)		
EO	5	115.70 (33.049)	74.01 (56.881)	−41.68 (28.554)		

Results are expressed as mean (standard deviation) for each group. C: control; CO: commercial fish oil; EO: Indonesian shortfin eel by-product oil. * One-way ANOVA Test; ** *p*-value < 0.005.

## Data Availability

The data presented in this study are available upon request from the corresponding author.

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
