# Peer review of "Effects of Indonesian Shortfin Eel (Anguilla bicolor) By-Product Oil Supplementation on HOMA-IR and Lipid Profile in Obese Male Wistar Rats"

_nutrients, 2023, doi:10.3390/nu15183904_

Round 1

Reviewer 1 Report (New Reviewer)

The paper describes an intervention in obese rats with an oil derived from the Indonesian Shortfin eel by-product investigating the effect of this oil  to glycemia, insulin resistance and lipid profile indices. For comparison the authors used a contol diet and a commercially available fish oil. 

Major comments

The methodology section is too extended and detailed in the abstract while at the same time the results are not presented thoroughly

Table 2.2. The sum of SFA+MUFA+PUFA is almost 100% for the Indonesian Shortfin Eel while it is ~ 75% for the commercial oil. Can the authors give an explanation for this ? Moreover, the authors should present a Table showing the daily amount of each fatty acid provided by the CO and EO.  

The preparation of the Indonesian shortfin eel by-product oil should be described in the methodology section. 

Tables 2.1, 3.1. The data are presented as Mean (Standard Error) in Table 3.1 and as Mean (Standard Deviation) in Table 2.1. Why ? 

Figure 3.1 is not needed since it shows the same data as in Table 3.1

Table 3.2. There are large differences between the standard errors of the lipid parameters in each group. I suspect that the data are not normally distributed and that the authors should use non parametric tests for their analysis.

Discussion. "In this study, administration of fish oil (commercial fish oil and Indonesian shortfin eel by-product oil) still showed the ability of the Omega-3 fatty acids contained therein to inhibit insulin resistance, proven by the increase in HOMA-IR levels in the two groups which were less than the control group." This is not supported by the data of Table 3.1

Discussion. "Even though our findings contradict the previous studies, we believe there is still potential in using eel waste (by-products) as a source of Omega-3 Fatty Acids." At the same time the same oil is rich in SFA. Do the authors believe that the overall fatty acid profile of this oil is still beneficial for insulin resistance ? 

Minor comments

1. Introduction. The authors could give more recent data on the epidemiology and treatment of diabetes. References 2,3 should be replaced by newer ones.

2. Table 2.1. Change KKal to Kcal  

3. 2.3. Oil Dose Determination: "1000 g of commercial fish oil contains 300 mg..." . You propably mean 1000 mg

4. 2.7. Ethical Consideration: "......and 5F as follows: (1) Freedom from hunger and thirst; (2) Freedom from discomfort; (3) Freedom from discomfort;...". Freedom from discomfort is repeated. 

The use of the English language is acceptable but there are several syntactical errors throughout the paper. They should be corrected with the help of a native English speaker. 

Author Response

Dear Reviewer 1,

We would like to thank you for your feedback on my manuscript entitled "Effects of Indonesian Shortfin Eel (Anguilla Bicolor) By-Product Oil Supplementation in HOMA-IR and Lipid Profile in Male Obese Wistar Rats" (Manuscript ID: nutrients-2559152).

We hope that the improvements we make meet your expectations. This manuscript improvement also considers the opinions and suggestions of other reviewers. We hope that this manuscript can continue to be processed.

Kind regards ,

Ginna Megawati

Faculty of Medicine

Universitas Padjadjaran Bandung-Indonesia

Reviewer 2 Report (New Reviewer)

The abstract generally does not need to have so much methodological detail. For example, the equation for calculating HOMA-IR, the chemistry behind the lipid assays, the specific type of statistical test used for data analysis. All of these elements can just be described in the Methods section.

Can the authors please justify using Mead’s equation to determine sample size, rather than the more common and accurate Power calculations? The authors should at least perform a post-hoc power analysis given the relatively small sample size of the study.

More explanation is needed for the authors statement in the Methods sections that “the results were considered as outliers” (line 117). Is the determination of a result as an outlier made based on a statistical calculation or just the opinion of the analyst?

Based on the omega-3-based normalization approach, the EO group required 2-3 times as much volume of oil per day as the CO group. The potential confounding effects of this should be discussed.

The fact that blood was sampled by different methods for the baseline and post-treatment measurements could impact the measured outcomes. For example, most of the lipid measures in Table 3.2 appear to have been affected by saline in the control (C group).  The authors should comment on the possibility that some effect is due to different sampling methods.

“Freedom from discomfort” appears to be listed twice on line 210.

On line 216, the following statement refers to the incorrect table. This should refer to table 2.1. “The mean value of fasting plasma insulin and fasting plasma glucose were presented in Table 4.13”

Author Response

Dear Reviewer 2,

We would like to thank you for your feedback on my manuscript entitled "Effects of Indonesian Shortfin Eel (Anguilla Bicolor) By-Product Oil Supplementation in HOMA-IR and Lipid Profile in Male Obese Wistar Rats" (Manuscript ID: nutrients-2559152).

We hope that the improvements we make meet your expectations. This manuscript improvement also considers the opinions and suggestions of other reviewers. We hope that this manuscript can continue to be processed.

Kind regards ,

Ginna Megawati

Faculty of Medicine

Universitas Padjadjaran Bandung-Indonesia

This manuscript is a resubmission of an earlier submission. The following is a list of the peer review reports and author responses from that submission.

Round 1

Reviewer 1 Report

The study investigated the effect of Indonesian shortfin eel by-product oil (source of omega-3 PUFA) compared to water and commercial fish oil on HOMA-IR levels in obese male Wistar rats. The study showed no statistically significant differences on levels of HOMA-IR among the groups after 4 weeks of intervention. The design of the study could be improved, and the manuscript needs additional data. Sections of the manuscript must be improved.

Specific comments:
Line 19: Add omega-3 polyunsaturated fatty acids
Line 22: Indicate the mechanisms for estimating HOMA-IR
Line 23-24: Add omega-3 polyunsaturated fatty acids
Line 24: add brief rationale for the sentence "possibly preventing insulin resistance"
Line 26: Because is beginning of the sentence, write twenty-seven instead of 27
Line 26: Use "." instead of "," for the cutoff of the Lee index
Line 27: In the Methods section, the first group is supplemented with water instead of Nacl. Please, clarify. If needed, add concentration of NaCl solution.
Line 28-29: Please, clarify meaning of p-values. Are they for normality test?
Other comment for the Abstract section: Add briefly the parameters analyzed and methods performed
Line 42-44: Please, add references
Line 48: Please, add example(s) of health agencies and add corresponding reference(s)
Line 52-53: Add type of lifestyle modifications analyzed and add the reference of meta-analysis mentioned
Line 54-56: Indicate examples of diseases
Line 56-57: Brief description of mechanisms may be included
Line 55: Add omega-3 polyunsaturated fatty acids
Line 55: Add omega-3 polyunsaturated fatty acids
Line 66-68: Can the HOMA-IR cut-off be applied to rats?
Line 70: Add omega-3 polyunsaturated fatty acids
Line 75: Add omega-3 polyunsaturated fatty acids
Line 69-71: Add references
Line 93: Why the rats had initial age between 0-12 weeks? This is large variation
Line 93: Provide rationale for using of this rat model and males
Line 94: Provide details of PT Biofarma (city, country)
Line 99: Add reference for Lee index and cut-off for definition of obesity
Line 100: Provide details of the method used for allocation of animals into the 3 groups
Line 100: Provide rationale for comparison to water and the commercial fish oil
Line 100: Use "g" instead of "gr"
Line 100: Describe determination of length of the body
Line 101: Water or NaCl solution?
Line 101: Provide details of commercial company for fish oil
Line 109: Define extreme levels of HOMA-IR
Line 111: Provide rationale for using of 4-weeks period of intervention. This may be too short to observe effects of the oils
Line 113: Why blood was taken from heart at the end of the study instead of tail vein?
Table 1: Use "g" instead of "gram" and "body weight" instead of "weight"
Table 1: Add baseline values of length of the body, Lee index and HOMA-IR
Line 125: Provide reference
Line 128-129: Provide reference an a brief description of the method
Line 131: Provide reference
Line 133: Polyunsaturated fatty acid was already abbreviated
Table 2: What the % is referred to?
Table 2: Provide full fatty acid composition of oils
Table 2: Please, also clarify composition of commercial fish oil
Line 141-142: Was administration of oils performed daily?
Line 143-144: Provide rationale for dose used and commercial fish oil used
Line 154-155: Provide details of commercial companies for glucose and insulin tests
Line 157-159: Explain limitations
Line 163: Which criteria was used for identification of outliers?
Line 166: Which test was used for testing homogeneity of variances?
Table 3 and Figure 1 show same data. Choose one of them. Was the ANOVA test applied to the Differences? Add values of body weight, glucose and insulin at the end of the study.
Line 206: Describe and provide references of contradictions
Line 208 and 212: Add omega-3 polyunsaturated fatty acids
Line 217: Provide reference(s)
Line 230: Provide reference(s)
Line 246: Explain controversies and add references.
Line 247-252: Provide references
Line 253-254: Provide reference
Line 254-257: Provide reference
Line 257-260: Provide reference
Line 261-264: Provide reference
Line 288, 293, 301, 304: Add omega-3 polyunsaturated fatty acids
Line 292: HOMA-IR was already abbreviated
Line 295: Indicate the other groups used for comparison in references 6 and 7
Line 304-305: No statistically significant difference was observed between the C group and the CO group.

Author Response

Dear Reviewer 1,

We would like to thank you for your feedback on my manuscript entitled "Indonesian Shortfin Eel By-Product Oil Supplementation Effects on HOMA-IR Levels in Male Obese Wistar Rats" (Manuscript ID: nutrients-2232614).

We hope that the improvements we make meet your expectations. We've created this response in tabular form to make it easier to read and understand the changes. This manuscript improvement also considers the opinions and suggestions of other reviewers. We hope that this manuscript can continue to be processed.

Kind regards

Ginna Megawati

Reviewer 2 Report

In the manuscript, the author described Indonesian Shortfin Eel By-Product Oil Supplementation Effects on HOMA-IR Levels in Male Obese Wistar Rats. They found that supplementing indonesian shortfin eel by-product oil in the rat obesity model did not decrease HOMA-IR level. The manuscript contains some issues, therefore, I suggest major reversion before consideration for publication in this journal.

1. The reference format is not uniform, such as in line 68, line 69, and line 135. The references format needs to be corrected, such as references 1, 2, and 46.

2. HOMA-IR level is calculated based on fasting plasma glucose and insulin levels. The insulin and glucose levels in plasma measured after interventions should be added to the manuscript.

3. Table 3 shows no difference in the HOMA-IR levels of the three groups of rats before and after the intervention. The difference in the EO group tends to increase, which may be due to the difference in the HOMA-IR levels of the three groups of rats before the intervention, and the HOMA-IR levels of the three groups are very close after the intervention.

4. The full name of HOMA-IR appears in the manuscript for the first time.

5. There are extra spaces in line 103 of the manuscript.

Author Response

Dear Reviewer 2,

We would like to thank you for your feedback on my manuscript entitled "Indonesian Shortfin Eel By-Product Oil Supplementation Effects on HOMA-IR Levels in Male Obese Wistar Rats" (Manuscript ID: nutrients-2232614).

We hope that the improvements we make meet your expectations. We've created this response in tabular form to make it easier to read and understand the changes. This manuscript improvement also considers the opinions and suggestions of other reviewers. We hope that this manuscript can continue to be processed.

Kind regards

Ginna Megawati

Round 2

Reviewer 1 Report

The revised version of the manuscript shows some improvements compared to the previous version. However, the manuscript needs additional data and additional experiments (for example, other parameters related to obesity and insulin resistance, and incorporation of fatty acids in blood and/or tissues) and some parts of the manuscript need to be clarified (for example, the fatty acid profile of commercial fish oil and the references used). Writing may be more clear and concise.

Some comments:
Line 60: Reference 4 is a review, not a meta-analysis. Reference of the meta-analysis must be included.
Line 74-76: References 18, 19 and 20 show data in human populations. Reference 20 do not use HOMA-IR. Is HOMA-IR and the mentioned cut-off validated in rats? Provide appropriate reference(s). Why oral glucose tolerance tests and insulin tolerance tests were not performed?
Line 81: Reference 23 is not a study performed in an experimental animal model.
Line 114-16: Primary reference and validation studies on Lee index must be cited. Reference 32 of the manuscript shows that body mass index correlates with metabolic alterations, not Lee index. If available, weight of adipose tissue may also be included in the table of characteristics of rats.
Line 160-162: Are both oils in triglyceride form?
Table 1: Use “.” Instead of “,” in values of insulin and glucose.
Table 1: Log-transformed data should be presented as geometric mean and corresponding measure of dispersion.
Table 2: Use “.” instead of “,”
Table 2: Fatty acid profile of commercial fish oil must be fully characterised.
Line 162: According to Table 2, Indonesian shortfin eel by-product oil contains 23.13%. Please, clarify.
HOMA-IR values (before and after) are presented in Table 1 and Table 3. Line 321-325: Writing of these sentences should be revised.
Caloric intake (diet plus supplement) during period of intervention should be included.

Author Response

Dear Reviewer 1,

We would like to thank you for your valuable feedback on my manuscript entitled "Indonesian Shortfin Eel By-Product Oil Supplementation Effects on HOMA-IR Levels in Male Obese Wistar Rats" (Manuscript ID: nutrients-2232614).

We hope that the improvements we make meet your expectations. We hope that this manuscript can continue to be processed.

Kind regards

Ginna Megawati

Reviewer 2 Report

No more comments 

Author Response

Dear Reviewer 2,

We would like to thank you for your valuable feedback on my manuscript entitled "Indonesian Shortfin Eel By-Product Oil Supplementation Effects on HOMA-IR Levels in Male Obese Wistar Rats" (Manuscript ID: nutrients-2232614).

We hope that the improvements we make meet your expectations. We hope that this manuscript can continue to be processed.

Kind regards

Ginna Megawati